# No Ergogenic Effect of Caffeine or Sodium Bicarbonate on Resistance Exercise Performance: A Double-Blind Crossover Study with Sex-Based Analysis

**DOI:** 10.3390/sports13120427

**Published:** 2025-12-03

**Authors:** Melissa L. A. Williams, Catherine Mary Evelyn Barrett, Ethan Lawson, Colin P. Major, Ashley Sandra May Shea, Karlie Squires, Megan Squires, Reza Zare, Katie M. Heinrich, David George Behm

**Affiliations:** 1School of Human Kinetics and Recreation, Memorial University of Newfoundland, St. John’s, NL A1C 5S7, Canada; mwilliams20@mun.ca (M.L.A.W.); cmebarrett@mun.ca (C.M.E.B.); elawson@mun.ca (E.L.); cpmajor@mun.ca (C.P.M.); asmshea@mun.ca (A.S.M.S.); kisquires@mun.ca (K.S.); mcsquires@mun.ca (M.S.); 2SRH Campus Hamburg, SRH University of Applied Sciences, 20095 Hamburg, Germany; 3Department of Kinesiology, College of Health and Human Sciences, Kansas State University, Manhattan, KS 66506, USA; kmhphd@ksu.edu; 4Department of Research and Evaluation, The Phoenix, Denver, CO 80205, USA

**Keywords:** chest press, knee extension, supplements, ergogenic, resistance training

## Abstract

Caffeine, a nervous system stimulant, and sodium bicarbonate, a metabolic buffer, have been shown to improve performance in high-intensity, particularly aerobic, exercises. This randomized, double-blind, placebo-controlled, crossover study compared the acute effects of caffeine and sodium bicarbonate on chest press (CP) and knee extension (KE) performance. Twelve resistance-trained young adults (seven females) completed three testing sessions during which they consumed caffeine (0.003 g/kg of body weight), sodium bicarbonate (0.3 g/kg of body weight), or placebo at 120, 90, and 60 min prior to testing. Testing consisted of six sets of CP and KE for as many repetitions as possible at 70% of the participants’ 1-repetition maximum load. A 60 s recovery between sets, and 2 min recovery was allocated between exercises. Blood pressure and blood lactate were recorded pre-, mid-, and post-test. Heart rate, ratings of perceived exertion (RPE), repetitions, and electromyography (EMG) were recorded for every set. No significant differences were found under any condition for RPE, EMG, and KE repetitions. A main effect for groups revealed was increased CP repetitions with sodium bicarbonate (7.42; 95%CI: 6.8–7.9) versus caffeine (6.7; 95%CI: 6.1–7.3) and control (7.1; 95%CI: 6.4–7.6) conditions. However, post hoc analysis did not achieve significance. Diastolic blood pressure was significantly (*p* = 0.03) greater with caffeine (79.2 mmHg; 95%CI: 74.6–83.7) than with sodium bicarbonate (72.7 mmHg; 95%CI: 67.5–77.9) and control (74.5 mmHg; 95%CI: 71.7–77.3). Females had significantly lower blood lactate measurements, higher CP repetitions, and lower heart rates, despite similar resting heart rates between the sexes. Caffeine (0.003 g/kg) or sodium bicarbonate (0.3 g/kg) did not provide acute ergogenic effects on CP or KE strength endurance (six sets of CP and KE at 70% 1-repetition maximum load) performance in young adult women and men.

## 1. Introduction

Supplements, including caffeine and sodium bicarbonate consumption, are commonly used to enhance exercise performance [1]. Since caffeine’s removal from the World Anti-Doping Agency’s prohibited substances list in 2004, there is a renewed interest in caffeine supplementation [2]. Caffeine is consumed globally daily in many forms (e.g., drinks, edibles, prescriptions) to reduce fatigue and perceived effort while increasing alertness and vigor through the antagonization of adenosine receptors in the central nervous system [3,4].

Muscular strength and endurance are affected by various neural factors, including enhanced motor unit recruitment, synchronization, rate coding, and neuromuscular inhibition [5,6]. Caffeine-induced adenosine receptor antagonization can also increase motor unit recruitment, hence caffeine’s popularity as an ergogenic supplement for high-intensity physical tasks across sports modalities [7]. In addition to neural effects, caffeine can also increase mobilization, and myofibrillar sensitivity of calcium ions [8]. Caffeine is considered beneficial for aerobic sports, due to ergogenic benefits of decreased pain perception and increased endurance. However few investigations have explicitly focused on resistance exercise [2,6,7,9,10] and muscle contractile properties [11]. The magnitude of caffeine’s ergogenic effects range from positive to no effect to negative effects on strength (effect sizes ranged from 0.16 to 0.20) and endurance (effect sizes ranged from 0.28 to 0.38) performance [7,12]. A recent umbrella meta-analysis of nine meta-analyses reported that caffeine increased muscle strength and endurance primarily with male participants who consumed the caffeine typically 60 min prior to testing [13].

Further, research has proposed intracellular water content as an indicator for muscle strength in certain populations [14], likely due to the effect of fluid volume on energy metabolism [15]. As such, studies report increased intracellular hydration in resistance-trained males and females [16,17]. Contrary to the general belief that caffeine is dehydrating due to diuretic effects, studies show no such effects during exercise when hydration was measured via fluid retention and urine output [18,19]. Regarding caffeine dosage, it has been established that 3–9 mg/kg of body weight may elicit an exercise performance enhancement; however, researchers have noted that gaps in the literature exist concerning the effect of lower caffeine doses on short-burst anaerobic exercise [20,21].

Whereas caffeine impacts the central nervous system and muscle calcium sensitivity, sodium bicarbonate can enhance high-intensity exercise or sports performance by acting as a metabolic acidic buffer within the muscle [22,23]. Exercise-induced acidosis can induce muscular fatigue when the rate of hydrogen ion (H+) production exceeds the rate of removal [24]. The use of a buffering agent such as sodium bicarbonate can help to counterbalance H+ acidosis, thus reducing fatigue [24]. Studies have shown that sodium bicarbonate can act as both an ergogenic and ergolytic substance (i.e., side effects of gastrointestinal upset can adversely affect performance) [22]. A position stand by the International Society of Sports Nutrition [9] states that sodium bicarbonate supplementation can improve muscle endurance and exercise performance for both men and women (optimal dose of 0.3 g/kg taken 60–180 min before exercise).

The bicarbonate buffering capacity has implications for the gastric system, specifically the stomach and duodenum by neutralizing gastric acid [22]. Participants that have ingested sodium bicarbonate have experienced gastrointestinal upset, nausea, diarrhea, and vomiting [22]. Many factors are known to affect the efficacy of supplementation in improving exercise outcomes, such as dosage, timing, and training status [23]. Therefore, dosing is critical when administering sodium bicarbonate for ergogenic effects. The most commonly utilized dosage of sodium bicarbonate in the exercise science literature is 0.3 g/kg of body weight [22].

Disparities exist within the literature on the effects of caffeine and sodium bicarbonate supplementation on each sex [23,25]. Research into nutritional supplements for women has been scarce and needs more attention [9,10,23]. The common exclusion of female participants from sports science research has been routinely justified based on menstrual hormonal changes [26]. This approach has led to the current inequality between male and female data in the field of supplement research for enhancing exercise performance [9,10,23].

As there are fewer studies directly comparing caffeine and sodium bicarbonate acute supplementation effects, especially with women, this exploratory study aimed to assess and compare the efficacy of an acute dose of caffeine (neuromuscular stimulant) and sodium bicarbonate (peripheral metabolic buffer) as an ergogenic aid for knee extension (KE) and chest press (CP) strength endurance. This study included both women and men, thus contributing more female data to an underrepresented field.

## 2. Methods

### 2.1. Participants

This study included data from 12 participants (mean age: 21.7 ± 1.4 years, range: 20–25 years). The required sample size was determined using an a priori statistical analysis (G*power version 3.1.9.2, Dusseldorf, Germany). Based on pre- to post-test force data from prior similar studies [4,27,28], the mean differences between two dependent means (matched pairs) (f-test: test family) were used to determine that approximately 12 participants were needed to achieve an alpha of 0.05, an effect size of 0.5 (moderate magnitude), and a power of 0.8. However, readers should be cautioned that this participant power estimation was based on a single test (e.g., pre- to post-testing). With a series of 2-way ANOVAs investigating a large cadre of tests, the sample may be considered small and thus the findings could be considered more exploratory rather than strongly confirmatory.

There were seven females (mean height: 164.9 ± 9.6 cm; mean body mass: 66.4 ± 12.6 kg) and five males (mean height: 176.3 ± 2.5 cm; mean body mass: 86.4 ± 12.2 kg). The average KE 1 RM was 80.2 ± 20.3 kg (51.0–113.4 kg) for females and 129.5 ± 23.1 kg (102.1–162.0 kg) for males. The average CP 1 RM for females was 47.0 ± 14.9 kg (24.9–70.3 kg) and 110.2 ± 24.5 kg (83.9–147.4 kg) for males. Women’s KE and CP 1 RM were 61.9% and 42.6% of the corresponding values for men. Convenience and snowball sampling were used to recruit participants, and posters were placed throughout the Memorial University of Newfoundland, St. John’s campus. The inclusion criteria were apparently healthy (self-reported), physically active males and females between ages 18–40 with regular (minimum three times a week) resistance training experience over the past 6 months, who regularly consumed (at least once a day) caffeine. The exclusion criteria were people below 18 years old or over 40 years old; people who were not fluent in English to decrease the possibility of confusion with instructions that could impact protocols; people with physical and/or cognitive impairment that would prevent physical activity inclusion; and people currently using other ergogenic supplements.

All participants were informed of the procedures and potential risks before giving their written informed consent to participate. This study was performed in accordance with the Declaration of Helsinki. The research protocol was pre-registered with Clinical Trials (ClinicalTrials.gov ID: NCT06714331) and was approved by the Interdisciplinary Committee on Ethics in Human Research (ICEHR) of the Memorial University of Newfoundland on 11 April 2024 under protocol number #20241611.

### 2.2. Experimental Design

The study employed a randomized (for the supplement conditions), double-blind, placebo-controlled, crossover study design that examined the acute effects of caffeine and sodium bicarbonate on performance using an isoinertial knee extension resistance device and Olympic bar bench press. All randomization-based allocation sequences were concealed until the commencement of the interventions. The two exercises (CP and KE) were chosen as they involve large muscle masses for the upper and lower limbs, respectively. Parameters included number of repetitions until failure with a 12-repetition maximum (12 RM) load, rating of perceived exertion (RPE), electromyography (EMG) activity, blood lactate, blood pressure, heart rate, intracellular and extracellular body water, and extracellular body water versus total body water measurements. Participants visited the School of Human Kinetics and Recreation applied neuromuscular research laboratory on four separate occasions to complete familiarization. There were three supplement conditions: (1) placebo, (2) caffeine, and (3) sodium bicarbonate, in a randomized order (www.thewordfinder.com/random-letter-generator, 28 March 2024). Supplements were prepared by a researcher who was not present at the testing protocol. This researcher kept a private record of each participant’s supplement sequence to blind participants and researchers during the testing protocols. During each testing protocol, participants completed CP and KE in a randomized order (www.thewordfinder.com/random-letter-generator). Testing was performed approximately one week apart at approximately the same time of day to allow for recovery and supplement washout and avoid possible diurnal variations in performance. There were no test timing differences for female participants since research does not show significant effects of the menstrual cycle on acute strength or power performance [29,30,31,32,33,34]. Participants were instructed to avoid vigorous exercise and refrain from consuming alcohol, caffeine, and other stimulants 24 h before each experimental trial. Figure 1 describes the experimental design.

### 2.3. Familiarization

Participants completed the Physical Activity Readiness Questionnaire for Everyone (PARQ+) [35]. Participants’ anthropometric measurements were taken, including height via a wall-mounted stadiometer and body composition via an InBody machine (InBody Co., Ltd., Seoul, Republic of Korea). Participants were introduced to the 60 beats per minute (bpm) baseline which was used during the testing protocol to maintain a cadence of 1 s for both the concentric and eccentric muscle actions. Participants were also introduced to the RPE scale, a modified Borg CR10 scale ranging from 1 to 10 with adjectives to help participants identify their exertion level. Participants then underwent 1 RM testing for KE and CP exercises to determine their maximal load for one repetition. Each participant’s 1 RM was used to calculate their 12 RM load for the subsequent experiment trials. This calculation was completed using the National Strength and Conditioning Association Training Load Chart; 70% of the 1 RM load was used as the estimated 12 RM load [36].

### 2.4. Body Composition and Water Content

InBody bioelectric impedance (Ottawa, ON, Canada) analysis was completed prior to supplement consumption and after the testing protocol. Pre- and post-test intracellular water (kg), extracellular water (kg), and extracellular water/total body water (ECW:TBW) ratio data were recorded from a standing (erect) position. Body weight was recorded during the pre-test (GE body weight scale). Reliability coefficients for InBody testing values are reported by the company to be 0.99.

### 2.5. Supplement Protocol

The supplement protocol began 120 min before the testing protocol and immediately following the initial InBody analysis. All three supplement conditions followed the same protocol: three opaque bottles containing 300 mL of fluid were consumed each session (900 mL of fluid total per session). The first 300 mL bottle was consumed two hours (120 min) in advance of the testing protocol; the second 300 mL, 90 min in advance; and the third 300 mL, 60 min in advance. All bottles were consumed as quickly as the participant was able to tolerate. All supplements and control conditions were prepared with water and Mio water flavoring (Pittsburgh, PA, USA, Kraft Heinz) in all three bottles to ensure relatively similar texture and flavor. For the caffeine condition, commercial grade caffeine was used (Atelier Evia, Repentigny, QC, Canada). The dose of caffeine was 0.003 g/kg of body weight, since this dose is suggested to elicit ergogenic effects without the possible side effects present in higher doses [37]. This caffeine dose was present only in the third of three 300 mL bottles (60 min in advance), with the first two 300 mL bottles containing only placebo. Food-grade sodium bicarbonate was used (Arm & Hammer, Cheyenne, WY, USA). The concentration dose of sodium bicarbonate was 0.3 g/kg of body weight, as is commonly used in the literature to explore ergogenic effects with minimal gastrointestinal side effects [4,38]. The sodium bicarbonate was consumed at 120, 90, and 60 min prior to testing to minimize possible gastrointestinal distress.

### 2.6. 12 RM Testing Protocol

Following supplement consumption, participants performed an aerobic warm up of five minutes of stationary cycling at 60–70 rpm at 1 kilopond resistance on a Velotron ergometer (Velotron RacerMate, Seattle, WA, USA). Participants then did a resistance training warm up of 8–10 repetitions of KE (Cybex isoinertial machine) and CP (Olympic bar) at 50% of the weight (random order allocation). In order to induce significant fatigue [39,40] and, based on a prior supplement study [41], participants were instructed to complete as many repetitions as possible against a 12 RM (approximately 70% of 1 RM according to National Strength and Conditioning Association) [36] load for each set (6 sets for each exercise with the two exercises performed in a randomized order) until they could no longer complete a repetition without assistance or maintenance of the prescribed cadence. Multiple-set protocols at 70–85% 1 RM (≈8–12 RM range) are common in resistance-exercise research for assessment of muscular endurance and supplement effects since they stress glycolytic pathways and accumulate fatigue, which are ideal conditions to detect ergogenic benefits [42]. The number of repetitions successfully completed during each set was recorded. Based on prior research from this lab and others [40,41,43,44,45], the participants had a 60 s rest period between each set which is consistent with muscular endurance training guidelines from American College of Sports Medicine [42]. They recommend short rest periods (30–90 s) to maximize metabolic stress and fatigue accumulation. Based on similar inter-set rest periods [39,41,44,46], following completion of 6 sets of the first exercise (CP or KE), participants rested for 120 s before completing the same procedure on the next exercise (CP or KE).

### 2.7. Electromyography (EMG)

Since caffeine is considered a central nervous system stimulant, EMG was recorded to determine any possible changes in muscle activation [3,4]. EMG electrode placement was conducted by the same researcher with one year of EMG placement experience to ensure consistency and validity. They were placed 3 cm apart at the mid-belly of the lateral head of the triceps brachii (mid-point between acromion process to the olecranon process) and 3 cm apart at the mid-belly of the rectus femoris (mid-point between the anterior superior iliac spine and the superior edge of the patella). Ground electrodes were placed on the acromion process for the upper limb and on the fibular head for the lower limb. A thorough skin preparation was executed prior to electrode placement, including shaving and removing dead epithelial cells with an abrasive pad, then cleansing the aforementioned areas with an isopropyl alcohol swab. EMG was collected using a Biopac (Biopac System Inc., DA 100: analog–digital converter MP150WSW; Holliston, MA, USA) data acquisition system at a sample rate of 2000 Hz [impedance = 2 MΩ, common mode rejection ratio > 110 dB min (50/60 Hz), noise > 5 μV]. A bandpass filter (10–500 Hz) was applied prior to digital conversion.

The mean amplitude of the root mean square (RMS) EMG was monitored during the middle 500 ms period of the 1 s concentric phase for both the KE and CP. The mean amplitude of the RMS EMG was normalized to the highest pre-test value and reported as a percentage.

### 2.8. Blood Lactate, Heart Rate (HR), and Ratings of Perceived Exertion (RPE)

Blood lactate levels were measured with the Edge Blood Lactate Analyzer (Atwood, CA, USA) three times, once prior to exercise, once between the two exercises, and immediately after the testing protocol was completed. Reliability is reported to be high for this device (0.97). The participant’s HR (bpm: Polar H10 heart rate monitor with chest and wrist straps, Polar Electro, Kempele, Finland), blood pressure (Omron Health Care upper arm cuff), and RPE (modified Borg CR 1–10 scale) were collected immediately after the completion of each set. The descriptive Borg CR 1–10 RPE scale was placed in the field of view of participants.

### 2.9. Statistical Analysis

Statistical analyses were calculated using SPSS software (Version 28.0, SPSS, Inc., Chicago, IL, USA). Shapiro–Wilk tests of normality were conducted for all dependent variables. Significance was defined as *p* < 0.05. If the assumption of sphericity was violated, the Greenhouse−Geiser correction was employed. A series of repeated measures ANOVAs with sex as a between group factor were conducted with Bonferroni post hoc tests corrected for multiple comparisons (α-value divided by the number of analyses on the dependent variable) to detect significant main effect differences and identify the significant interactions. The repeated measures ANOVAs included 3 conditions × 2 times (pre- and post-test) for blood lactate, systolic, and diastolic blood pressure and ECW/TBW. A 3 conditions × 6 sets repeated ANOVA was employed for KE and CP repetitions to failure, RPE, rectus femoris, and triceps brachii EMG, while a 3 conditions × 7 times (resting heart rate and 6 sets) was implemented for KE and CP heart rates. Partial Eta-squared (η_p_^2^) values are reported for main effects and overall interactions representing small (0.01 ≤ η_p_^2^ < 0.06), medium (0.06 ≤ η_p_^2^ < 0.14), and large (η_p_^2^ ≥ 0.14) magnitudes of change (from SPSS-tutorials, 2022). Observed power (OP) was also reported for the main effects and interactions.

## 3. Results

A total of 15 individuals fulfilled the study inclusion criteria. However, only 12 individuals completed the study. Details of enrollment, exclusion, randomization of supplement conditions, and final number of participants analyzed in this study are provided in Figure 2.

### 3.1. Knee Extension (KE) and Chest Press (CP) Repetitions

There were no significant main effects for supplements or any interactions for KE repetitions. A significant main effect for supplements (F_(2,20)_ = 4.35, *p* = 0.027, η_p_^2^: 0.303, OP: 0.686) was evident with CP repetitions. Although, the sodium bicarbonate condition exhibited the greatest number of CP repetitions (7.42; 95%CI: 6.8–7.9), the Bonferroni post hoc analysis indicated non-significant *p* values of 0.15 and 0.18 when compared to the caffeine (6.7; 95%CI: 6.1–7.3) and control (7.1; 95%CI: 6.4–7.6) conditions, respectively.

### 3.2. EMG and RPE

There were no significant main effects or interactions evident for triceps brachii EMG or rectus femoris EMG. There were no significant main effects for supplements or any interactions for KE or CP RPE.

### 3.3. Diastolic Blood Pressure

A main effect for the supplement condition (F_(2,20)_ = 4.17, *p* = 0.03, η_p_^2^: 0.294, OP: 0.66) indicated that the caffeine condition diastolic blood pressure (79.2 mmHg; 95%CI: 74.6–83.7) was significantly greater than for the sodium bicarbonate (72.7 mmHg; 95%CI: 67.5–77.9) and control (74.5 mmHg; 95%CI: 71.7–77.3) conditions.

### 3.4. Extracellular Water/Total Body Water (ECW/TBW)

A significant time–sex interaction (F_(1,9)_ = 5.46, *p* = 0.04, η_p_^2^: 0.378, OP: 0.55) revealed that both males (0.368–0.371) and females (0.372–0.377) experienced pre- to post-test increases in ECW/TBW. A significant supplements–time interaction (F_(2,18)_ = 16.49, *p* = 0.001, η_p_^2^: 0.647, OP: 0.977) showed that all three conditions demonstrated an increase in the ECW/TBW ratio with no significant differences between conditions.

### 3.5. Main Effects for Time and Sets

Significant main effects for time (pre- to post-test) demonstrated increases in ECW/TBW, blood lactate, and systolic blood pressure (Table 1). Repetitions generally decreased from the first to the subsequent sets for both KE and CP whereas KE RPE increased from the first to subsequent sets (Table 2). Heart rate significantly increased when compared to the resting heart rate with a plateau in subsequent sets for KE and only minor but significant differences with CP (Table 3).

### 3.6. Sex Differences

Table 4 illustrates significant main effects for sex with lower blood lactates in females, versus higher CP repetitions for females. A significant sets–sex interaction (F_(6,60)_ = 2.66, *p* = 0.023, η_p_^2^: 0.210, OP: 0.826) for CP HR showed that while resting heart rates were similar between the sexes, females experienced lower heart rates during each of the six sets.

## 4. Discussion

This exploratory study aimed to investigate the ergogenic effects of an acute dose of caffeine and sodium bicarbonate for KE and CP resistance training. Acute strength endurance performance as measured by increased repetitions and decreased RPE was not enhanced in young adult women and men. The major findings were that 300 mL of solution containing caffeine (0.003 g/kg solution), 900 mL of solution containing sodium bicarbonate (0.3 g/kg), or dextrose (control) ingested at 120, 90, and 60 min before the testing protocol did not enhance KE repetitions, rectus femoris or triceps brachii activation (EMG), or RPE. There was a large magnitude effect size, a significant main effect for the group performing a greater number of CP repetitions after ingesting a sodium bicarbonate solution; however, the post hoc analysis did not achieve significance. Caffeine ingestion did elevate diastolic blood pressure. The interventions in general (main effects for time or sets) induced large magnitude increases in blood lactate, RPE, ECW/TBW, systolic blood pressure, and decreased repetitions. Females produced lower blood lactate levels and a greater number of CP repetitions. However, with only 7 female and 5 male participants, the analysis of sex differences should be considered exploratory.

### 4.1. Knee Extension (KE) and Chest Press (CP) Repetitions

We found that acute caffeine supplementation did not enhance KE or CP muscle endurance as measured by the number of repetitions successfully performed. The literature on the acute ergogenic effects of caffeine for increased repetitions is inconsistent, especially when comparing the upper and lower body. Contrary to our results, prior research supports acute and chronic training ergogenic effects of caffeine on upper body resistance exercise/training [6,37]. A recent meta-analysis of the literature concluded that there is a significant increase in the number of CP repetitions performed until failure in those who ingested caffeine compared to placebo control [47]. Regarding lower body performance, the literature is contradictory, with most studies reporting increased repetitions during caffeine supplementation [6,37,47,48]. However, these ergogenic results are most commonly reported in squat and leg press exercises [37]. In those studies that have also tested knee extension, significant increases in repetitions compared to placebo control have been found with doses of 0.005 g/kg, while no improvements are reported at a 0.002 g/kg dose [6,37].

This suggests that the lack of significant increases in our results may be due to our caffeine dosage of 0.003 g/kg. Prior studies have reported ergogenic effects with as low as 0.002 g/kg with specific lower body exercises (i.e., leg press and squats). The International Society of Sports Nutrition recommends a minimum dosage of 0.002 g/kg [7]. Doses of 0.004 g/kg to 0.006 g/kg are reported for ergogenic effects on upper body performance and 0.005 g/kg to 0.006 g/kg for significance in knee extension exercise [37]. Thus, our exploratory findings agree with much of the current literature that there is no difference in upper and lower body endurance with caffeine doses below 0.004 g/kg [8].

In accordance with our exploratory findings, many previous studies show no significant effect of sodium bicarbonate on the number of CP [49,50] or KE repetitions [38,51]. However, sodium bicarbonate has contributed to a significant increase in the time to fatigue in cycling to exhaustion tests [52,53]. This contrast can likely be attributed to differences in exercise duration and intensity. Shorter, high-intensity exhaustion tests rely on relatively more energy from anaerobic metabolism (greater accumulation of H+) compared to traditional resistance training, facilitating earlier fatigue [53]. It is proposed that the sodium bicarbonate ingestion can inhibit pH decrease in the muscle and increase the removal rate of H+ [53]. As a result of the brief duration of the CP and KE sets (estimated 12 RM from 70% 1 RM) with the 60 s recovery between sets, the accumulation of H+ ions (acidosis) would not be as high as some other high-intensity cycling or running protocols [53] and thus the role of sodium bicarbonate as a buffer [22] was not as consequential in the present study.

Prior studies have shown a significant acute increase in muscular endurance as a result of the placebo effect [49,54]. Further, in a recent study, a significant increase in upper body strength was only found when supplementation was compared to a no-placebo control versus placebo control [48]. It is possible that if participants in this study believed they were consuming caffeine during the control session, they might perform better, thus skewing their baseline measures. Researchers have suggested asking participants to indicate which trial they believe to be caffeine, to distinguish between caffeine and placebo effects [6]. This was not performed in this study and should be considered as a methodological limitation.

### 4.2. Rating of Perceived Exertion (RPE)

Our finding that caffeine had no effect on RPE aligns with most of the literature that also reports no reduction in RPE [3,4,6,28,48,55,56,57]. Caffeine is a popular ergogenic aid for aerobic exercises, where it has been reported to provide a 5% decrease in RPE, but few resistance training studies have found similar results [58]. In many of these aerobic studies, RPE is lower during the prolonged activity, but is unchanged at the end of the exercise [58,59,60]. It is possible that since RPE measurements in this study occurred at the end of each set (i.e., following the activity) instead of during the sets, no reduction in effort was perceived [6].

No significant effect of sodium bicarbonate on RPE was apparent in our protocol of acute bouts of CP and KE resistance training; this concurs with the current literature [6,27,50,61]. In contrast, Marriott et al. [62] found that sodium bicarbonate significantly reduced RPE in a Yo-Yo intermittent recovery test (20 m running test at progressively increasing speeds) after an upper body workout. The Yo-Yo test, described as high-intensity intermittent exercise, produced greater heart rates than our resistance training protocol, once again identifying intensity as a factor that differs between the present study and studies that found ergogenic effects for sodium bicarbonate [62].

### 4.3. Electromyography (EMG)

We found no effect of either caffeine or sodium bicarbonate on EMG amplitude in either CP or KE exercises, which aligns with a few studies that have implemented short-term, high-intensity exercise or isometrics [38,63,64]. While no apparent studies have investigated the effect of caffeine on EMG during CP exercises, Trevino et al. [65] found no effect on biceps brachii EMG amplitude during isometric elbow flexion.

### 4.4. Blood Pressure

Both resistance training and caffeine have been shown to lead to increased blood pressure [6]. This large magnitude increase in blood pressure is supported by our study and is an important note of caution for individuals with high blood pressure. Caffeine stimulation of the central nervous system could increase blood pressure by inhibiting adenosine receptors, leading to increased neurotransmitter release, such as dopamine, acetylcholine, epinephrine, and norepinephrine [47,66,67].

### 4.5. Extracellular Water/Total Body Water (ECW/TBW)

ECW/TBW was elevated after the resistance training intervention for all conditions. This response is attributed to physiological responses to stress, such as the release of stress hormones like cortisol and inflammatory responses, which can promote water retention. As muscles enlarge in response to resistance exercise, they require greater amounts of intracellular water, which is initiated by an increase in extracellular water [16].

### 4.6. Sex Differences

The sex differences seen in this study were lower blood lactate levels, higher CP repetitions, and lower working heart rates in females. Lower blood lactate levels in females have been observed in a variety of both aerobic and resistance training studies [68,69,70]. In a study of high-intensity (75% of 1 RM), short-rest (1 min) resistance training, males demonstrated significantly higher blood lactate levels. Further, lactate has a strong positive correlation with total work done in kilograms [68], which is in accord with the higher levels of total work for males in the present study.

Regarding higher repetitions of CP in females, in tests of isometric endurance females have demonstrated less fatigability than males [71]. However, a narrative review of sex differences reported that sex differences in muscle endurance during non-isometric programs are mixed [72]. Specifically, the review cited that evidence of sex differences in muscle fatigability measured through repetitions to failure at relative loads, as in our study design, is lacking [72]. It is important to note that as seen in our study, sex differences in fatigability do appear to be task specific and may be influenced by factors such as the muscle group assessed [71,72].

Females tend to have higher resting heart rates [73], however in the present study we found no difference in resting heart rate between sexes. In fact, females in our study had lower working heart rates than males. Two non-sex difference related mechanisms behind lower heart rates are lower heart rates in more highly trained individuals (inclusion criteria of at least 6 months of resistance training in the present study), and lower heart rates due to a lower relative workload [73,74].

Sex differences in both CP repetitions and heart rate could be theoretically related to the relative load determined in this study through the 1 RM test. Resistance training experience level has been suggested in some literature to impact the accuracy of 1 RM testing based on individuals producing greater 1 RM results with more testing sessions [75]. There is also greater variability in the maximum repetitions performed at any percentage-based training load with lower percentages of 1 RM load [72]. However, a recent systematic review has found that 1 RM testing generally has between good and excellent test–retest reliability regardless of resistance training experience and sex [48].

An important consideration to make when interpreting our results is sample size. A greater sample size could have strengthened statistical power. We did compute statistical power for a single statistical test but with only 7 females and 5 males, the multiple tests involving repeated measures 2-way ANOVAs and statistical sex interaction findings should be considered exploratory. It was difficult to attract participants to volunteer for a familiarization session and three 3 h experimental sessions, which included the possibility of gastrointestinal distress. Additionally, participant self-reporting may have affected results if participants did not fully follow pre-test instructions (e.g., no vigorous activity or alcohol 24 h before testing). Furthermore, participants were not required to change their dietary habits prior to lab visits. As the participants were habituated to coffee intake, the ergogenic effects could have been masked, with higher caffeine dosages required to produce an ergogenic effect. Research regarding supplementation on a fed versus fasted state has elucidated that a 0.003 g/kg dose is only ergogenic in a fasted state [6]. Finally, since we asked participants to do as many repetitions as possible for each set, it is possible they rated their RPE immediately after each set as corresponding with their level of effort.

Despite clearly defining the inclusion criteria as participants regularly performing resistance training (minimum three times a week) for the past 6 months, it is still vague. This description can include those who may have had years of extensive training experience, while also including individuals who may have only begun training 6 months ago affecting the homogeneity of the sample. For example, the range within and between male CP 1 RM (*n* = 5, Range = 185–325 lbs) and female CP 1 RM was considerable (*n* = 8, Range = 55–155 lbs).

Our study, however, had numerous strengths, including its double-blind, crossover, placebo-controlled design. Furthermore, the inclusion of a washout period between testing protocols and a 24 h abstinence from stimulants ensured that our results accurately reflect the effects of an acute dosage of each supplement with no interactions. Lastly, this study expands the literature on the sex-based differences in resistance training performance following supplement ingestion.

## 5. Conclusions

This study observed no significant acute ergogenic effects for muscle endurance, EMG, or RPE during KE and CP exercises from caffeine at a 0.003 g/kg dosage or sodium bicarbonate supplementation at a 0.3 g/kg dosage when ingested 60–120 min prior to exercise in young adult women and men. While sodium bicarbonate supplementation seemed to increase CP repetitions, these interaction findings were non-significant. This study adds to the currently contradictory supplementation literature by concluding that 0.003 g/kg may be too minimal a caffeine dosage to elicit acute strength endurance improvements during resistance training. The results and interpretations are specific to the volume and intensity of load used in this study with results possibly differing if alternative volumes and intensities were tested. Further research should include more participants and different age populations, investigate sex differences further, and focus solely on upper or lower body protocols to determine ergogenic dosages for each.

## Figures and Tables

**Figure 1 sports-13-00427-f001:**
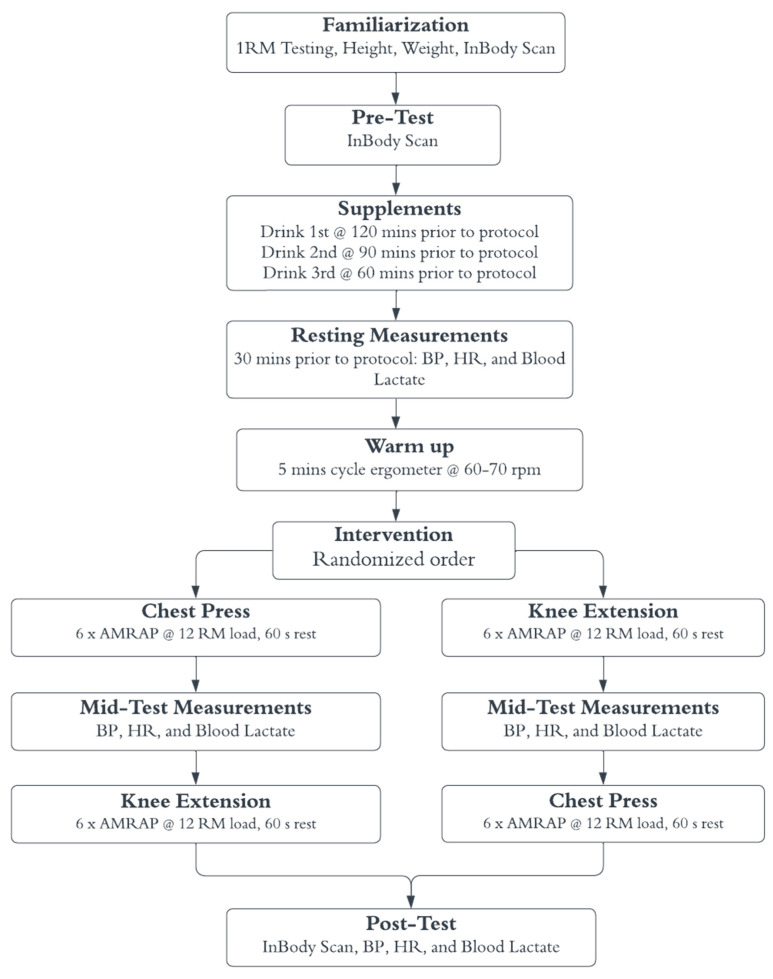
Experimental Design. 1 RM: 1-repetition maximum testing, BP: blood pressure, HR: heart rate, AMRAP: as many repetitions as possible (until failure), 12 RM: calculated 12-repetition maximum load used as the testing load.

**Figure 2 sports-13-00427-f002:**
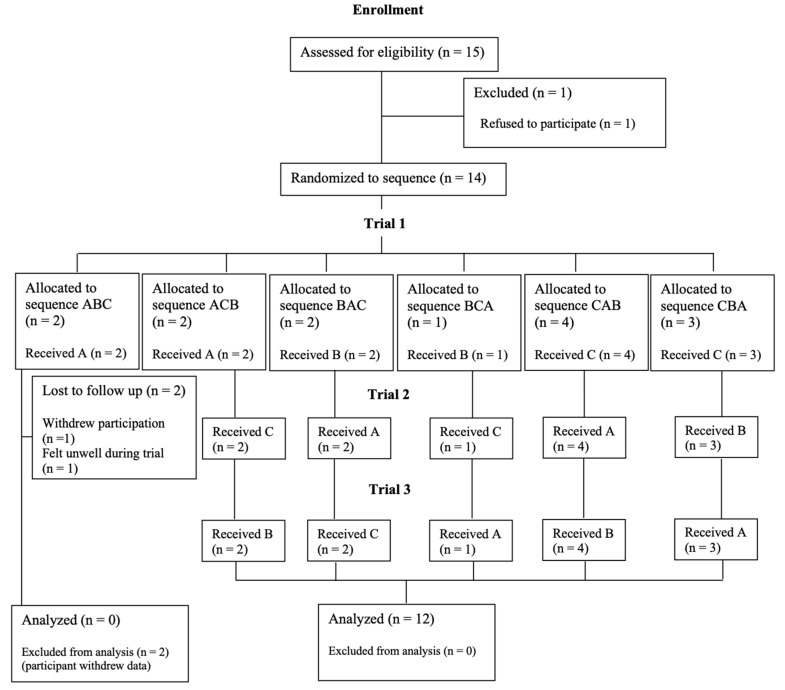
CONSORT flow diagram of the three supplement conditions. A = placebo; B = sodium bicarbonate; C = caffeine.

**Table 1 sports-13-00427-t001:** Physiological responses to exercise: Pre- and post-test comparisons with effect sizes and statistical power.

	Pre-Test Means (95% CI)	Post-Test Means (95% CI)	F Values	*p* Value and Effect Size	Observed Power
ECW/TBW	0.370 (0.364–0.376)	0.374 (0.369–0.380)	114.54	<0.001 η_p_^2^: 0.927	1.00
Blood Lactate (mg/dL)	37.13 (26.5–47.7)	130.11 (112.5–147.6)	63.15	<0.001 η_p_^2^: 0.875	1.00
Systolic Blood Pressure (mmHg)	115.71 (110.8–120.5)	132.04 (127.3–136.7)	24.87	<0.001 η_p_^2^: 0.713	1.00

Note. ECW/TBW: extracellular water/total body water.

**Table 2 sports-13-00427-t002:** Repetitions, rating of perceived exertion (RPE), and rectus femoris EMG across sets: Means, 95% confidence intervals, and main effects.

	Set 1	Set 2	Set 3	Set 4	Set 5	Set 6	F	*p*	Power
Knee extension repetitions	13.46 (11.5–15.3)	8.71 (7.6–9.8)	7.2 (6.1–8.2)	6.5 (5.6–7.4)	5.6 (4.7–6.5)	6.03 (5.1–7.0)	99.1	<0.001 η_p_^2^: 0.908	1.00
Sig different from other sets	2–6	1, 3–6	1, 2, 4–6	1–3, 5	1–4	1–3			
Knee extension RPE	7.67 (6.6–8.7)	8.39 (7.6–9.1)	8.75 (8.1–9.3)	9.03 (8.5–9.5)	9.31 (8.8–9.7)	9.45 (9.0–9.8)	20.1	<0.001 η_p_^2^: 0.668	1.00
Sig	2–6	1, 5, 6	1, 6	1, 5	1, 2, 4	1–3			
Rectus femoris EMG (final/1st rep)*100	30.46 (1.3–62.2)	18.18 (2.6–33.6)	10.84 (5.5–27.2)	6.08 (8.1–20.3)	6.79 (4.6–18.2)	6.96 (2.2–16.1)	2.7	0.08 η_p_^2^: 0.217	0.78
Sig	NS	NS	NS	NS	NS	NS			
Chest press repetitions	15.04 (13.6–16.4)	7.81 (6.8–8.8)	5.62 (4.7–6.5)	4.80 (4.2–5.4)	4.51 (3.8–5.1)	4.58 (3.7–5.4)	118.1	<0.001 η_p_^2^: 0.922	1.00
Sig	2–6	1, 3–6	1, 2	1, 2	1, 2	1, 2			
Chest press RPE	7.46 (6.4–8.4)	8.02 (7.2–8.8)	8.57 (7.9–9.1)	8.88 (8.3–9.4)	9.05 (8.5–9.5)	9.3 (8.7–9.8)	21.3	<0.001 η_p_^2^: 0.681	1.00
Sig different from other sets	3–6	3–6	1, 2, 4–6	1–3	1–3	1–3			

Note. EMG: electromyography, RPE: rating of perceived exertion, rep: repetition, Sig: significantly different from another set(s), NS: no significance.

**Table 3 sports-13-00427-t003:** Heart rate responses across sets during knee extension and chest press: Means, 95% confidence intervals, and main effects.

	RHR	Set 1	Set 2	Set 3	Set 4	Set 5	Set 6	F	*p*	Power
KE HR	64.79 (61.3–68.3)	142.62 (120.2–165.1)	135.69 (115.9–155.4)	137.38 (119.9–154.8)	137.06 (121.7–152.4)	135.31 (119.7–150.9)	138.61 (123.5–153.7)	64.28	<0.001 η_p_^2^: 0.865	1.00
Sig	1–6									
CP HR	64.79 (61.3–68.3)	142.07 (123.6–160.4)	134.40 (116.8–151.9)	134.12 (119.5–148.7)	127.47 (111.6–143.3)	130.78 (115.1–146.4)	134.5 (120.0–149.1)	74.84	<0.001 η_p_^2^: 0.887	1.00
Sig	1–6	4–5	4		1, 2, 6	1	4			

Note. KE: knee extension, CP: chest press, HR: heart rate, RHR: resting heart rate, Sig: significantly different from another set(s).

**Table 4 sports-13-00427-t004:** Comparison of blood lactate and chest press repetitions between males and females.

	Males	Females	F	*p*	Power
Blood lactate	98.15 (84.1–112.1)	77.38 (64.59–90.17)	6.13	0.035 η_p_^2^: 0.405	0.598
Chest press repetitions	4.3 (3.5–5.1)	9.8 (9.1–10.5)	138.49	<0.001 η_p_^2^: 0.933	1.00

## Data Availability

Data are available upon request.

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
