# Peer review of "No Ergogenic Effect of Caffeine or Sodium Bicarbonate on Resistance Exercise Performance: A Double-Blind Crossover Study with Sex-Based Analysis"

_sports, 2025, doi:10.3390/sports13120427_

Round 1
Reviewer 1 Report
Comments and Suggestions for Authors
A great study to fill in the lack of female participants in the literature on this particular topic.
The design and procedures were flawless with potential confounding variables controlled for.
All sections, particularly the Discussion was well organized and easily readable and interpretable.
Author Response
Reviewer #1
A great study to fill in the lack of female participants in the literature on this particular topic.
The design and procedures were flawless with potential confounding variables controlled for.
All sections, particularly the Discussion was well organized and easily readable and interpretable.
Authors’ response: We would like to thank the reviewer for their time and effort. Your positive comments are much appreciated.
Reviewer 2 Report
Comments and Suggestions for Authors
Dear authors,
I preface this review by stating that I am not an expert on supplementation, and therefore some of my comments might be misplaced. Overall, the article is very well written and structured, with solid methodological foundations. I do believe there is room for improvement, hence my specific comments below.
Title
- The title is very clear and informative. While it could benefit from adding information regarding the training status of the participants, I feel it would make the title overly long, so I guess it is a piece of information that can be left to the abstract.
Abstract
- It is conceivable that certain supplements may have chronic effects, even if their acute effects are negligible. Therefore, when mentioning “the effects of caffeine and sodium bicarbonate”, please add “acute”. To avoid repeating myself, please add “acute” to all appropriate locations throughout the manuscript.
- The information of 120 to 60 minutes prior is unclear – it seems that each participant could decide when to ingest the supplement within that time window. Later, in the manuscript, readers realize this is not what happened. Please rephrase to better align the abstract with the full text.
- Otherwise, the abstract is very well written. Some of my concerns and questions are better addressed in the full manuscript, not here. Therefore, I left the comments in the subsequent sections of my review.
- However, I think the conclusion of the abstract needs to be better contextualized. In particular, the performance was mostly based on strength endurance (i.e., many sets * many repetitions), and the results could have been different in other contexts (e.g., few sets, few repetitions, higher loads).
- The same for the take home message: not RT performance in general, but specifically strength endurance.
Introduction
- The introduction could benefit from some streamlining/synthesis. However, since the journal does not impose word limits, I leave the decision to the authors.
- There is, however, an imbalance, whereas the extent of discussion surrounding caffeine is way greater than that devoted to sodium bicarbonate.
- Also, throughout the introduction, please insert the term “acute” where appropriate. Although “ergogenic” is commonly associated with acute effects, this is not necessarily the case. Therefore, please stress that the effects being referred to (as well as the effects that this manuscript aimed to explore) are acute.
- Please provide some rationale for the choice of chest press and knee extension specifically, either in the introduction, or in the methods. I guess the authors wanted to check two exercises involving large muscle masses, one for the upper limbs and the other for the lower limbs, but it would be better to state that explicitly.
- This is clearly an exploratory study: (i) small sample; (ii) too many statistical tests to be run. Therefore, there is no need to state a formal hypothesis. Indeed, I believe it is ill-advised to state formal hypotheses in this context. Even if they are theoretically sound, this manuscript does not possess sufficient power to address these effects in a confirmatory manner.
Methods
- Increasingly, randomized trials should be pre-registered, and some journals demand that. You mention the ClinicalTrials.gov registration, but please state explicitly whether it was pre-registered or post-registered (the site allows both).
- If possible, start the methods with a “study design” or “overview” section, synthesizing the main points of the methods. Some journals demand it, and I feel it helps bringing the reader to an understanding of the gist of the study’s mechanics. Still, this is optional, so I leave the decision to the authors.
- 12 participants seem a small number, therefore better configuring this manuscript as an exploratory rather than confirmatory study. As such, maybe justify this sample as being both purposeful and by convenience, and assume the lack of calculation of a priori statistical power. Later, when discussing the findings, perhaps use more tentative language, such as “suggest”, to denote that these findings should be interpreted with caution.
- Sample size calculation: I do not feel it is appropriate for this study. Even if the authors keep this here, such calculations are for a single statistical test (i.e., a primary outcome, although this outcome may combine several tested variables). Because the authors have performed multiple statistical tests, this calculation is basically useless for most practical purposes. If assuming the study as being exploratory, this step is not required.
- The alpha size is problematic: given the large number of statistical tests, keeping p value at 0.05 means there is a large chance of obtaining false positives. The alternative would be implementing family-wise error rate corrections, but that would increase false negatives. The bottom line is: please try being extra careful when presenting and interpreting the findings, avoiding over-reliance on p values.
- The a priori effect size of 0.5 is pretty large – was this based in studies with comparable sample (age, training status) and interventions (e.g., number of sets * repetitions)? If not, a smaller a priori ES would be more appropriate.
- The power calculation is misleading, given my previous arguments. This is, in fact, an underpowered sample, because of a potentially inflated ES and – more importantly – the large number of statistical tests that were performed (the power calculation is for a single statistical test). Please consider using instead the argument of a purposeful and convenience sample, and frame the study as being exploratory.
- The participants have a very narrow age range – this should be reflected in the conclusions, including the conclusions of the abstract, to avoid generalization to populations of very different ages.
- My previous comment is strengthened by the number of outcomes and, consequently, the number of statistical tests that were performed. On one hand, the small sample means “non-significant” findings may simply be false negatives due to poor statistical power. On the other hand, the number of statistical tests could result in “significant” results by chance. In a nutshell, I would suggest not putting too much weight on the p values when discussing the findings and consider them more as suggestions of interesting stuff that should be studied in future research using an appropriately powered sample for a specific, primary outcome.
- The participants were previously habituated to coffee intake, which could have masked the ergogenic effects. Habituation could mean that higher doses of caffeine would be required to produce an ergogenic effect. For the discussion, it would be important to state that the ergogenic effects of certain substances may require progressively higher doses as habituation sets in. I’m leaving out sodium bicarbonate here, because I don’t believe many people use it regularly, at least not in the sense that a large part of the general population drinks coffee or tea daily (therefore being exposed to caffeine).
- When mentioning randomization, please explicitly refer whether allocation sequence was concealed until the beginning of the interventions. I’m guessing it was, but better to make it explicit (important for future RoB 2 assessments, for example).
- Overall great study design, and wash-out period that was more than enough for these purposes.
- The bottles were opaque, which is great, but were all three supplements (caffeine, sodium bicarbonate, and placebo) similar in texture and flavour, to avoid potential placebo (or nocebo) effects? I feel that the drinks probably had different flavours (and possibly textures as well), which could have annulled the blinding. Please refer to this in the methods and, if I’m right, please report as a limitation later in the study.
- The testing clearly focused on strength endurance (6 sets * multiple repetitions). Could the results have been different for other configurations? For example, is it possible that the results would have been different if the participants were tested for a lower number of sets and repetitions, perhaps with increased loads? I think this design is ok but should not be extended or generalized to other types of resistance training. Therefore, this should be acknowledged throughout the discussion and conclusion.
- Electromyography: because its application and testing are highly dependent on observer/assessor/examiner experience, please report who performed these assessments and what their experience level was.
- For testing devices, please provide known reliability values, and preferably their typical errors. For example, different blood lactate analysers perform better at different concentrations.
- Statistical analyses: consider not focusing too much on p values, given the simultaneous likelihood of false positives (number of tests performed) and false negatives (small sample for the number of tests performed). Unclear how these two potential effects will interact, so it’s advisable to also focus on the descriptive statistics and effect sizes (regardless of the result being “significant” or “non-significant”.
Results
- The lack of significance of some results may have resulted from poor statistical power. Therefore, I think the authors did well in highlighting some trends, even if failing to achieve statistical power.
- However, for the results that came back “significant”, please add sentences to qualify them and tone-down their interpretation. Namely, given the number of statistical tests that were performed, “significant” findings may have been reached purely by chance.
- Given the constraints that I highlighted previously, consider using expressions such as “suggested there were no significant main effects (…)”.
- The tables are missing legends for some abbreviations (e.g., CI).
Discussion
- Given my previous concerns regarding the sample size and number of statistical tests that were run, please slightly re-word some passages to provide a more tentative account of the results (e.g., “hinted at”, “suggestive of”).
- Also considering that, I would frame the study as exploratory and avoid mentioning formal hypotheses.
- In the same vein, contextualize the findings to the specific type of resistance training protocol applied here, with its many sets and repetitions, and clarify that the effects could have been different for distinct RT protocols.
- In previous comments, I left additional suggestions for information that should be reported in the limitations.
- Otherwise, the discussion is very well-written and balanced.
Conclusions
- Again, please emphasize this applies mostly to strength endurance protocols and to young adults ~20 years of age.
Author Response
Reviewer #2
I preface this review by stating that I am not an expert on supplementation, and therefore some of my comments might be misplaced. Overall, the article is very well written and structured, with solid methodological foundations. I do believe there is room for improvement, hence my specific comments below.
Authors’ response: We would like to thank the reviewers for their time and effort. Your constructive comments have improved the manuscript and are much appreciated.
Title
The title is very clear and informative. While it could benefit from adding information regarding the training status of the participants, I feel it would make the title overly long, so I guess it is a piece of information that can be left to the abstract.
Authors’ response: We agree, the population is typically mentioned in the title but with the detail already included, the title length would be cumbersome.
Abstract
It is conceivable that certain supplements may have chronic effects, even if their acute effects are negligible. Therefore, when mentioning “the effects of caffeine and sodium bicarbonate”, please add “acute”. To avoid repeating myself, please add “acute” to all appropriate locations throughout the manuscript.
Authors’ response: Done.
The information of 120 to 60 minutes prior is unclear – it seems that each participant could decide when to ingest the supplement within that time window. Later, in the manuscript, readers realize this is not what happened. Please rephrase to better align the abstract with the full text.
Otherwise, the abstract is very well written. Some of my concerns and questions are better addressed in the full manuscript, not here. Therefore, I left the comments in the subsequent sections of my review.
Authors’ response: Sentence has been rephrased as follows:
Twelve resistance-trained young adults (seven females) completed three testing sessions during which they consumed caffeine (0.003g/kg body weight), sodium bicarbonate (0.3 g/kg body weight), or placebo at 120, 90, and 60 minutes prior to testing.
However, I think the conclusion of the abstract needs to be better contextualized. In particular, the performance was mostly based on strength endurance (i.e., many sets * many repetitions), and the results could have been different in other contexts (e.g., few sets, few repetitions, higher loads).
The same for the take home message: not RT performance in general, but specifically strength endurance.
Authors’ response: We have made the suggested revision as follows:
Caffeine (0.003g/kg) or sodium bicarbonate (0.3 g/kg) did not provide acute ergogenic effects on CP or KE strength endurance (6 sets of CP and KE at 70% 1-repetition maximum load) performance.
Introduction
The introduction could benefit from some streamlining/synthesis. However, since the journal does not impose word limits, I leave the decision to the authors.
Authors’ response: We have synthesized/integrated some of the sentences in the introduction to provide more clarity.
There is, however, an imbalance, whereas the extent of discussion surrounding caffeine is way greater than that devoted to sodium bicarbonate.
Authors’ response: We would respectfully point out that the introductory information on caffeine includes 3 paragraphs whereas the sodium bicarbonate is 2 paragraphs. Hence there is not a large difference in the background information of the two supplements.
Also, throughout the introduction, please insert the term “acute” where appropriate. Although “ergogenic” is commonly associated with acute effects, this is not necessarily the case. Therefore, please stress that the effects being referred to (as well as the effects that this manuscript aimed to explore) are acute.
Authors’ response: Done
Please provide some rationale for the choice of chest press and knee extension specifically, either in the introduction, or in the methods. I guess the authors wanted to check two exercises involving large muscle masses, one for the upper limbs and the other for the lower limbs, but it would be better to state that explicitly.
Authors’ response: As suggested we have added your explanation in the Experimental Design section.
The two exercises (chest press and knee extension) were chosen as they involve large muscle masses for the upper and lower limbs respectively.
This is clearly an exploratory study: (i) small sample; (ii) too many statistical tests to be run. Therefore, there is no need to state a formal hypothesis. Indeed, I believe it is ill-advised to state formal hypotheses in this context. Even if they are theoretically sound, this manuscript does not possess sufficient power to address these effects in a confirmatory manner.
Authors’ response: We have removed the hypothesis as suggested.
Methods
Increasingly, randomized trials should be pre-registered, and some journals demand that. You mention the ClinicalTrials.gov registration, but please state explicitly whether it was pre-registered or post-registered (the site allows both).
Authors’ response: Information added as suggested:
The research protocol was pre-registered with Clinical Trials (ClinicalTrials.gov ID: NCT06714331) and was approved by the Interdisciplinary Committee on Ethics in Human Research (ICEHR) of the Memorial University of Newfoundland on 11/04/2024 under protocol number #20241611.
If possible, start the methods with a “study design” or “overview” section, synthesizing the main points of the methods. Some journals demand it, and I feel it helps bringing the reader to an understanding of the gist of the study’s mechanics. Still, this is optional, so I leave the decision to the authors.
Authors’ response: Our 2.2 Experimental Design section provides a summary of the protocol for the readers.
12 participants seem a small number, therefore better configuring this manuscript as an exploratory rather than confirmatory study. As such, maybe justify this sample as being both purposeful and by convenience, and assume the lack of calculation of a priori statistical power. Later, when discussing the findings, perhaps use more tentative language, such as “suggest”, to denote that these findings should be interpreted with caution.
Authors’ response: We have emphasized the exploratory nature of this study as suggested and have added magnitude effect size descriptions to the text for clearer context for the readers. We have also cautioned the readers about the statistical power calculations.
Sample size calculation: I do not feel it is appropriate for this study. Even if the authors keep this here, such calculations are for a single statistical test (i.e., a primary outcome, although this outcome may combine several tested variables). Because the authors have performed multiple statistical tests, this calculation is basically useless for most practical purposes. If assuming the study as being exploratory, this step is not required.
Authors’ response: We would like the readers to know that we did go thought the exercise of calculating a sample size. However, based on your comments we have added this cautionary note:
However, readers should be cautioned that this participant power estimation was based on a single test (e.g., pre- to post-testing). With a series of 2-way ANOVAs investigating a large cadre of tests, the sample may be considered small and thus the findings could be considered more exploratory rather than strongly confirmatory.
The alpha size is problematic: given the large number of statistical tests, keeping p value at 0.05 means there is a large chance of obtaining false positives. The alternative would be implementing family-wise error rate corrections, but that would increase false negatives. The bottom line is: please try being extra careful when presenting and interpreting the findings, avoiding over-reliance on p values.
Authors’ response: Point taken. We have altered the language in the manuscript to emphasize the exploratory nature of the results.
The a priori effect size of 0.5 is pretty large – was this based in studies with comparable sample (age, training status) and interventions (e.g., number of sets * repetitions)? If not, a smaller a priori ES would be more appropriate.
The power calculation is misleading, given my previous arguments. This is, in fact, an underpowered sample, because of a potentially inflated ES and – more importantly – the large number of statistical tests that were performed (the power calculation is for a single statistical test). Please consider using instead the argument of a purposeful and convenience sample, and frame the study as being exploratory.
Authors’ response: As mentioned we added the cautionary sentence re: this power calculation.
The participants have a very narrow age range – this should be reflected in the conclusions, including the conclusions of the abstract, to avoid generalization to populations of very different ages.
Authors’ response: We have added the population to the conclusions where appropriate. For example:
Abstract
Caffeine (0.003g/kg) or sodium bicarbonate (0.3 g/kg) did not provide acute ergogenic effects on CP or KE strength endurance (6 sets of CP and KE at 70% 1-repetition maximum load) performance in young adult women and men.
- Discussion
This study aimed to investigate the ergogenic effects of an acute dose of caffeine and sodium bicarbonate for KE and CP resistance training. Contrary to our hypothesis, acute exercise performance as measured by increased repetitions and decreased RPE was not enhanced in young adult women and men.
- Conclusions
This study observed no significant acute ergogenic effects for muscle endurance, EMG, or RPE during KE and CP exercises from caffeine at a 0.003g/kg dosage or sodium bicarbonate supplementation at a 0.3g/kg dosage when ingested 60-120 minutes prior to exercise in young adult women and men. While sodium bicarbonate supplementation seemed to increase CP repetitions, these interaction findings were non-significant. This study adds to the currently contradictory supplementation literature by concluding that 0.003g/kg may be too minimal a caffeine dosage to elicit acute performance improvements during resistance training. Further research should include more participants, different age populations, investigate sex differences further, and focus solely on upper or lower body protocols to determine ergogenic dosages for each.
My previous comment is strengthened by the number of outcomes and, consequently, the number of statistical tests that were performed. On one hand, the small sample means “non-significant” findings may simply be false negatives due to poor statistical power. On the other hand, the number of statistical tests could result in “significant” results by chance. In a nutshell, I would suggest not putting too much weight on the p values when discussing the findings and consider them more as suggestions of interesting stuff that should be studied in future research using an appropriately powered sample for a specific, primary outcome.
Authors’ response: Point taken. We have made the suggested adjustment throughout the manuscript. Furthermore, we have also added the following addition to the discussion section:
An important consideration to make when interpreting our results is sample size. A greater sample size could have strengthened statistical power. We did compute statistical power for a single statistical test but with only 7 females and 5 males, the multiple tests involving repeated measures 2-way ANOVAs and statistical sex interaction findings should be considered exploratory.
The participants were previously habituated to coffee intake, which could have masked the ergogenic effects. Habituation could mean that higher doses of caffeine would be required to produce an ergogenic effect. For the discussion, it would be important to state that the ergogenic effects of certain substances may require progressively higher doses as habituation sets in. I’m leaving out sodium bicarbonate here, because I don’t believe many people use it regularly, at least not in the sense that a large part of the general population drinks coffee or tea daily (therefore being exposed to caffeine).
Authors’ response: Based on your intrepid observations, we have added the following information based on your suggestions:
As the participants were habituated to coffee intake, the ergogenic effects could have been masked, with higher caffeine dosages required to produce an ergogenic effect.
When mentioning randomization, please explicitly refer whether allocation sequence was concealed until the beginning of the interventions. I’m guessing it was, but better to make it explicit (important for future RoB 2 assessments, for example).
Authors’ response: In the Experimental Design section we added the suggested information as follows:
All randomization-based allocation sequences were concealed until the commencement of the interventions.
Overall great study design, and wash-out period that was more than enough for these purposes.
Authors’ response: Thanks for the positive response.
The bottles were opaque, which is great, but were all three supplements (caffeine, sodium bicarbonate, and placebo) similar in texture and flavour, to avoid potential placebo (or nocebo) effects? I feel that the drinks probably had different flavours (and possibly textures as well), which could have annulled the blinding. Please refer to this in the methods and, if I’m right, please report as a limitation later in the study.
Authors’ response: We attempted to achieve similar tastes and textures for the three conditions. We have integrated that information. The original manuscript had this information in two different places so it was probably not clear.
All supplement and control condition were prepared with water and Mio water flavouring (Pittsburgh, Pennsylvania, USA, Kraft Heinz) in all three bottles to mask the supplements and to ensure relatively similar texture and flavour.
The testing clearly focused on strength endurance (6 sets * multiple repetitions). Could the results have been different for other configurations? For example, is it possible that the results would have been different if the participants were tested for a lower number of sets and repetitions, perhaps with increased loads? I think this design is ok but should not be extended or generalized to other types of resistance training. Therefore, this should be acknowledged throughout the discussion and conclusion.
Authors’ response: Reviewer is correct, the results are specific to the volume and intensity of load used in this study. We have added this information to the conclusion as follows.
The results and interpretations are specific to the volume and intensity of load used in this study.
Electromyography: because its application and testing are highly dependent on observer/assessor/examiner experience, please report who performed these assessments and what their experience level was.
Authors’ response: We have added the following information based on your comment:
EMG electrode placement was conducted by the same researcher with one year of EMG placement experience to ensure consistency and validity.
For testing devices, please provide known reliability values, and preferably their typical errors. For example, different blood lactate analysers perform better at different concentrations.
Authors’ response: We have added reliability values for the measures as requested.
Statistical analyses: consider not focusing too much on p values, given the simultaneous likelihood of false positives (number of tests performed) and false negatives (small sample for the number of tests performed). Unclear how these two potential effects will interact, so it’s advisable to also focus on the descriptive statistics and effect sizes (regardless of the result being “significant” or “non-significant”.
Authors’ response: We have added information about effect sizes as suggested. For example:
There was a large magnitude effect size, significant main effect for group for performing a greater number of CP repetitions after ingesting a sodium bicarbonate solution, however, the post-hoc analysis did not achieve significance.
The interventions in general (main effects for time or sets) induced large magnitude increases in blood lactate, RPE, ECW/TBW, and systolic blood pressure, and decreased repetitions.
This large magnitude increase in blood pressure is supported by our study and is an important note of caution for individuals with high blood pressure.
Results
The lack of significance of some results may have resulted from poor statistical power. Therefore, I think the authors did well in highlighting some trends, even if failing to achieve statistical power.
However, for the results that came back “significant”, please add sentences to qualify them and tone-down their interpretation. Namely, given the number of statistical tests that were performed, “significant” findings may have been reached purely by chance.
Given the constraints that I highlighted previously, consider using expressions such as “suggested there were no significant main effects (…)”.
The tables are missing legends for some abbreviations (e.g., CI).
Authors’ response: We have added a number of cautions in the discussion regarding the exploratory nature of the findings and the implications of the low sample size.
Discussion
Given my previous concerns regarding the sample size and number of statistical tests that were run, please slightly re-word some passages to provide a more tentative account of the results (e.g., “hinted at”, “suggestive of”).
Also considering that, I would frame the study as exploratory and avoid mentioning formal hypotheses.
Authors’ response: Done. Formal hypotheses have also been removed.
In the same vein, contextualize the findings to the specific type of resistance training protocol applied here, with its many sets and repetitions, and clarify that the effects could have been different for distinct RT protocols.
Authors’ response: We have ensured that our findings are specific to muscle strength endurance and removed more general descriptions such as referring to improvements in exercise performance. For example, in the conclusion we have added this following information:
This study adds to the currently contradictory supplementation literature by concluding that 0.003g/kg may be too minimal a caffeine dosage to elicit acute strength endurance improvements during resistance training. The results and interpretations are specific to the volume and intensity of load used in this study with results possibly differing if alternative volumes and intensities were tested.
In previous comments, I left additional suggestions for information that should be reported in the limitations.
Otherwise, the discussion is very well-written and balanced.
Authors’ response: We have added your suggested revisions. Thank-you for the positive comment regarding our discussion.
Reviewer 3 Report
Comments and Suggestions for Authors
Thank you for inviting me to review this study
ABSTRACT
Line 29 "Some evidence showed" please rephrase
INTRO
Line 48 "many forms" add details between brackets
Line 64-66 if possible report effect sizes
Line 126-129 it would be better to report relative values instead of absolute
STATISTICAL ANALYSIS
The sample size is not sufficient for the proposed analysis. This needs to be highlighted in the limitations section
Line 342 "there was some evidence" please rephrase
Line 356-357 include effect size if possible
SEX DIFFERENCES
No sufficient sample to draw any conclusion about it. I'd suggest to remove this aim or to provide a clear indication about this study limitations
Author Response
Reviewer #3
Authors’ response: We would like to thank the reviewers for their time and effort. Your constructive comments have improved the manuscript and are much appreciated.
ABSTRACT
Line 29 "Some evidence showed" please rephrase
Authors’ response: Changed to:
A main effect for groups revealed increased CP repetitions with sodium bicarbonate (7.42; 95%CI: 6.8-7.9), versus caffeine (6.7; 95%CI: 6.1-7.3) and control (7.1; 95%CI: 6.4-7.6) conditions.
INTRO
Line 48 "many forms" add details between brackets
Authors’ response: We have added the requested information as follows:
Caffeine is consumed globally daily in many forms (e.g., drinks, edibles, prescriptions).
Line 64-66 if possible report effect sizes
Authors’ response: Effect sizes reported as requested
“…on strength (effect sizes ranged from 0.16 - 0.20) and endurance (effect sizes ranged from 0.28 - 0.38) performance”
Line 126-129 it would be better to report relative values instead of absolute
Authors’ response: We have added the relative values as follows:
Women’s KE and CP 1 RM were 61.9% and 42.6% of the men respectively.
STATISTICAL ANALYSIS
The sample size is not sufficient for the proposed analysis. This needs to be highlighted in the limitations section
Authors’ response: We have now stated multiple times that this study is exploratory and even though the initial statistical power analysis indicated 12 participants would be adequate, the number of tests and factors would decrease statistical power. For example:
However, readers should be cautioned that this participant power estimation was based on a single test (e.g., pre- to post-testing). With a series of 2-way ANOVAs investigating a large cadre of tests, the sample may be considered small and thus the findings could be considered more exploratory rather than strongly confirmatory.
However, with only 7 female and 5 male participants, the analysis of sex differences should be considered exploratory.
An important consideration to make when interpreting our results is sample size. A greater sample size could have strengthened statistical power. We did compute statistical power for a single statistical test but with only 7 females and 5 males, the multiple tests involving repeated measures 2-way ANOVAs and statistical sex interaction findings should be considered exploratory.
Line 342 "there was some evidence" please rephrase
Authors’ response: The sentence has been rephrased as follows:
There was a large magnitude effect size, significant main effect for group for performing a greater number of CP repetitions after ingesting a sodium bicarbonate solution, however, the post-hoc analysis did not achieve significance.
Line 356-357 include effect size if possible
Authors’ response: We have added effect sizes to the discussion section. For example:
There was a large magnitude effect size, significant main effect for group for performing a greater number of CP repetitions after ingesting a sodium bicarbonate solution, however, the post-hoc analysis did not achieve significance.
The interventions in general (main effects for time or sets) induced large magnitude increases in blood lactate, RPE, ECW/TBW, and systolic blood pressure, and decreased repetitions.
This large magnitude increase in blood pressure is supported by our study and is an important note of caution for individuals with high blood pressure.
SEX DIFFERENCES
No sufficient sample to draw any conclusion about it. I'd suggest to remove this aim or to provide a clear indication about this study limitations
Authors’ response: We have now stated multiple times that this study is exploratory and that the even though the statistical power analysis indicated 12 participants would be adequate, the number of tests and factors would decrease statistical power. For example:
However, readers should be cautioned that this participant power estimation was based on a single test (e.g., pre- to post-testing). With a series of 2-way ANOVAs investigating a large cadre of tests, the sample may be considered small and thus the findings could be considered more exploratory rather than strongly confirmatory.
However, with only 7 female and 5 male participants, the analysis of sex differences should be considered exploratory.
An important consideration to make when interpreting our results is sample size. A greater sample size could have strengthened statistical power. We did compute statistical power for a single statistical test but with only 7 females and 5 males, the multiple tests involving repeated measures 2-way ANOVAs and statistical sex interaction findings should be considered exploratory.
Round 2
Reviewer 3 Report
Comments and Suggestions for Authors
Good work